# Position: Significant impact of numerical precision in scientific machine learning

## Abstract

The machine learning community has focused on computational efficiency, often leveraging lower-precision formats such as FP16, instead of the standard FP32. In contrast, little attention has been paid to higher-precision formats, such as FP64, despite their critical role in scientific domains like materials science, where even small numerical differences can lead to significant inaccuracies in physicochemical properties. This need for high precision extends to the emerging field of *machine learning for scientific tasks*, yet it has not been thoroughly investigated. According to several studies and our toy experiment, models trained with FP32 show insufficient accuracy compared to those trained with FP64, indicating that higher precision is also crucial in scientific machine learning, as in traditional scientific computing. This precision issue limits the potential of scientific machine learning that can replace the traditional scientific computings in practical research. Our position paper not only highlights these precision-related issues but also recommends reporting comparisons between FP32 and FP64 results, encouraging the release of FP64 models. We believe that these efforts can enable machine learning to contribute meaningfully to the natural sciences, ensuring both scientific reliability and practical applicability.

## 1 Introduction

The rapid advancements in natural language processing (NLP) and computer vision (CV) in the field of machine learning (ML) have accelerated the broad application across various domains [56, 73, 83, 51, 81]. Specifically, **ML for scientific tasks**–*which has begun to resolve intellectually demanding problems in scientific fields*–has been highlighted across disciplines, opening new possibilities for scientific breakthroughs. In recognition of these breakthroughs, the 2024 Nobel Prize in Chemistry honored the contributions of scientific ML, highlighting innovations such as AlphaFold and RoseTTAFold [39, 4, 99]. These models transformed scientific research by rapidly delivering results that once required significant resources and time-consuming experiments or simulations. Building on these successes, scientific ML not only addresses traditional labor-intensive workflows but also finds hidden patterns within complex data, thereby providing human researchers with direct insights into novel discoveries across natural sciences [109, 67, 41, 117, 97, 116].

In the context of methodology, the development of scientific ML naturally follows the broader trends and paradigms of the ML research field. In the early stages of NLP and CV, most work focused on discriminative tasks (*e.g.*, named entity recognition and image classification) [107, 24] before gradually shifting to generative tasks (*e.g.*, machine translation and text-to-image generation) [18, 89]. Further, generative approaches have advanced sequentially, moving from variational autoencoders (VAEs) to generative adversarial networks (GANs), and more recently, to diffusion models [43, 32, 35, 95]. In a similar manner, numerous scientific domains have rapidly adopted the latest advances from

the ML community. For example, among various areas of bioinformatics, research on DNA sequence data initially leveraged discriminative models such as DeepVariant [78] and DeepSEA [122], and over time, the trends moved to generative models including ExpressionGAN [124] and Evo [71]. Similarly, material structure prediction in the field of materials and drug discovery has followed this trend from VAEs [85, 31, 55] and GANs [79, 42, 1] to diffusion models [36, 75, 118].

In parallel with these advances, the scaling law, one of the most recent paradigm in ML research, emphasizes performance enhancement by progressively increasing the size of models, training datasets, and computational resources [40, 94]. Building upon this idea of incremental scale expansion, researchers have successfully tested the approach of *bigger is better* across diverse fields, including NLP, CV, reinforcement learning, and time-series forecasting [119, 22, 34, 70, 92]. Following this pattern, scientific ML is also adopting this paradigm, and in fact, large models designed to address scientific tasks have already begun to appear [71, 120].

As these models grow larger and more complex, they require massive computational resources, presenting significant challenges for both training and inference processess. To address this, the lower numerical precision and quantization are a widely employed strategy, which helps reduce the computational expense [123, 65]. These approaches inevitably involve a trade-off between fidelity and resource efficiency, typically resulting in some accuracy degradation. To minimize such precision-related losses, techniques such as mixed precision training [65] and sophisticated quantization methods [7, 25, 58, 112] have been proposed, which allow researchers to conserve the original accuracy while achieving the advantages of reduced computational costs. Consequently, the ML community has accepted slight accuracy degradation as a natural trade-off for greater efficiency, thereby integrating these lower-precision techniques into real-world applications to balance performance and computational burden.

However, the tolerance for lower-precision techniques raises substantial concerns in the field of scientific computing. Scientific computing primarily aims to solve fundamental physics equations that are difficult to solve manually by simplifying or discretizing the inherently continuous and infinite real-world phenomena to make them computationally tractable. As a consequence, even tiny differences in numerical precision can lead to significant issues regarding the reliability of computational results. Our experimental findings demonstrate that *single precision's sensitivity to numerical deviations can substantially influence the accuracy of fundamental physical equations*. As a result of this high sensitivity, small numerical differences can cause significant changes in physicochemical properties, such as absorption coefficient, defect energies, or reaction pathway predictions, thereby reducing the reliability of results, especially when accurate predictions are crucial for critical decisions. One critical aspect is that these challenges related to numerical precision are not confined to traditional computational science, as ML models are increasingly being utilized in various studies to replace prevalent simulations. In other words, traditional computational science requires high precision, making it essential to verify whether FP32 produces valid results before using ML models, as numerical precision is key to maintaining reliability.

**In this position paper, we argue for the significant role of numerical precision in scientific ML research, emphasizing the need for evaluating and analyzing its impact on results derived from varying precision levels.** To this end, we first highlight real-world examples from established computational simulations where numerical precision directly impacts on their results. We then explain that the importance of numerical precision is not confined to traditional scientific computing alone but is also deeply related to ML applications in scientific domains. Specifically, we provide examples involving ML potential models and physics-informed neural networks (PINNs), which are actively studied in both ML and science domains, demonstrating the critical role of numerical precision in these areas [82, 44, 50]. Additionally, we explore the implications of large language models (LLMs) in scientific ML on precision-related considerations.

In conclusion, we present concrete recommendations for the ML community and potential research directions based on our discussions. We then provide alternative viewpoints to our position, offer responses, and conclude. Since the main role of ML in scientific research is to deepen understanding in traditional domains, the issues we raise must be rigorously examined. When relatively simple actions by ML researchers can remove barriers that hinder natural scientists from applying ML models, these measures become essential, not optional. As scientific machine learning is still in its early stages, we hope that thorough debate will help minimize trial-and-error in future research.

## 2 Importance of numerical precision in scientific computing

The main goal of scientific computing is solving complex physics equations through computational power, especially when manual solutions are impractical or nonexistent. Specifically, many-body problems including multiple object interactions demonstrate the necessity of high-performance computing. Accordingly, various computational methods have emerged to solve fundamental physics equations: molecular dynamics for Newton's Second Law, density functional theory (DFT) [38] for the Schrödinger equation, and the finite-difference time-domain (FDTD) [115] method for Maxwell's equations. Despite the algorithmic progress outlined above, the fidelity of these simulations is bounded by how continuous physical variables are encoded on digital hardware. Modern digital processors represent real numbers as finite-length bit strings, so continuous equations—ranging from $F = ma$ to the Schrödinger and Maxwell formulations—cannot be solved exactly. To bridge this gap, scientists adopt controllable approximations: reformulating the problem (*e.g.*, the Kohn–Sham equation [45]) or discretising time and space (*e.g.*, molecular dynamics). These methods remain trustworthy only when round-off error is tightly bounded, making double-precision arithmetic the *de-facto* compromise between cost and accuracy. For instance, Quantum ESPRESSO [29], a leading open-source DFT implementation, strictly enforces double precision throughout its code.

To demonstrate the precision's crucial role in scientific computing, we present examples showing how small numerical variations can significantly impact computational results, analyzing these effects in realistic research scenarios. Specifically, we illustrate the influence on materials research scenarios, thereby analyzing the implications and identifying the precise numerical accuracy-related challenges.

### 2.1 Impact on density functional theory simulation

Quantum mechanics, beginning with Planck's quantum hypothesis [76], revolutionized our understanding of microscopic phenomena. While exact calculations are only possible for simple systems like the hydrogen atom, the Kohn-Sham equation introduced DFT as an efficient approach for many-body electron problems. Using Python-based Simulations of Chemistry Framework (PySCF) [98], we performed geometry optimization calculations for water ($H_2O$) using both Hartree-Fock (HF) and DFT calculations with B3LYP functional and 6-311++G(d,p) basis set [2, 102, 13, 113].

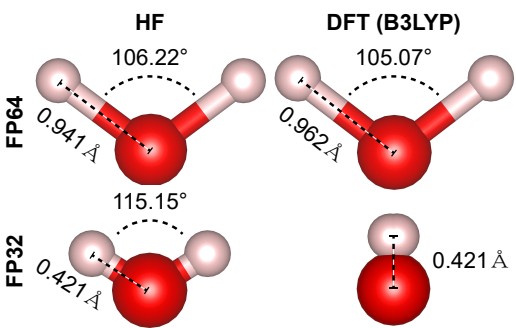

Figure 1: Geometry optimizations of a water molecule using FP64 (top) and FP32 (bottom) with HF (left) and DFT (right) methods. FP64 computations yield physically valid structures, whereas FP32 leads to unrealistic geometries.

Figure 1 shows the results of geometry-optimized water molecules obtained from HF and DFT calculations under FP32 and FP64 numerical precision conditions. When utilizing FP64, both HF and DFT calculations successfully converged within three optimization steps with satisfying the convergence criteria. Since DFT explicitly accounts for electron correlation effects [12], it is generally expected to provide more accurate results than HF, a trend that is also reflected in our findings. Comparing bond lengths, the reference [19] O-H bond length is 0.957 Å, while HF exhibits a deviation of 0.016 Å (1.7 % error), and DFT yields a smaller deviation of 0.005 Å (0.5 % error). Similarly, for the bond angle, HF deviates by $1.7°$ (0.7 % error) from the reference value of $104.52°$, whereas DFT shows a smaller deviation of $0.55°$ (0.5 % error). However, when using FP32, significant numerical instabilities arise, preventing the convergence of optimization steps. In the case of HF calculations, the gradient of hydrogen atoms stagnates between 0.2–0.4 Ha/Bohr, which is significantly above the desired convergence threshold of $10^{-6}$ Ha/Bohr. For DFT calculations, the issue becomes even more pronounced, as the gradient values rapidly diverge beyond $10^5$ Ha/Bohr, resulting in termination before reaching the maximum step. As a result, when using FP32, the HF calculation exhibits a substantial 50 % error, while the DFT calculation produces a molecular structure impossible to exist in reality, as illustrated in Figure 1. A detailed examination using the atomic coordinates is shown in Appendix B.1.

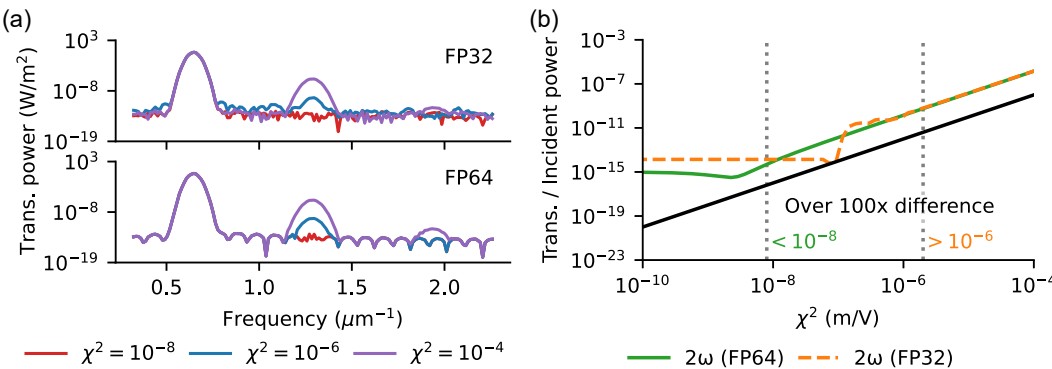

Figure 2: (a) Transmittance spectra comparison between FP32 (top) and FP64 (bottom) in Kerr media, showing FP32's failure to accurately model higher harmonics and low-power wave patterns below $10^{-10}$ W/m$^2$. (b) Computed second harmonic susceptibility shown in FP64 (green solid) and FP32 lines (orange dashed) compared to theoretical quadratic behavior (black). FP64 maintains accuracy to $10^{-8}$ m/V, while FP32 deviates above $10^{-6}$ m/V, making it unsuitable for typical nonlinear materials.

## 2.2 Impact on finite difference time domain simulation

Electromagnetism, established by Maxwell's equations [64, 63], provides the theoretical foundation for understanding electromagnetic waves. However, solving Maxwell's equations for complex phenomena is computationally challenging. To address this, FDTD discretizes Maxwell's equations in time and space. Using Meep [72], an open-source FDTD software, we investigated numerical precision effects on electromagnetic simulations, comparing FP32 and FP64 in nonlinear Kerr media simulations. We simulated a Kerr medium (refractive index= 1.65) excited by an electromagnetic wave source ($\lambda = 1.55\ \mu$m, $\Delta\lambda = 0.15\ \mu$m).

Figure 2 (a) presents the transmission spectrum of the nonlinear Kerr medium under FP32 and FP64 precision settings. From left to right, the spectral peaks correspond to the fundamental generation induced by the source, the second harmonic generation (SHG), and the third harmonic generation (THG). While the fundamental peak exhibits minimal differences between FP32 and FP64, notable discrepancies arise in the SHG and THG regions. Specifically, FP32 calculations display pronounced background signal instability and intensity variations in harmonic generation, which result from imprecise numerical computation. A particularly notable difference appears in the behavior of the background signal. In FP64 calculations, the background follows a well-defined periodic pattern governed by the electromagnetic wave, whereas in FP32, the background signal appears as unstructured Gaussian-like noise. This phenomenon indicates that the lack of numerical precision in FP32 significantly disrupts the accurate computation of low-intensity transmitted power, particularly for electromagnetic waves in the range of $10^{-11}$ W/m$^2$. These findings highlight the fundamental limitations of single precision in reliably capturing weak electromagnetic signals and nonlinear optical effects.

To further analyze the impact of numerical precision, we examined the relationship between second-order nonlinear susceptibility ($\chi^2$) and the transmittance-to-incident power ratio. As shown in Figure 2 (b), the black upward-sloping line represents a quadratic line, serving as a reference line indicating the expected computational trend of transmittance over incident power ratio as nonlinear susceptibility varies. Ideally, the computationally simulated values should align with this reference trend, maintaining the same slope. Comparing the results obtained from FP64 (green solid line) and FP32 (orange dashed line), we observe that as nonlinear susceptibility decreases beyond a certain threshold, the ratio begins to saturate. This saturation point effectively defines the lower bound of computational precision achievable under each numerical setting.

Specifically, for values of $\chi^2$ above $10^{-6}$, both FP64 and FP32 provide reliable computational precision. However, for values below this threshold, FP32 results begin to exhibit saturation, rendering further calculations meaningless due to the loss of numerical resolution. In contrast, FP64 maintains simulation accuracy down to approximately $10^{-8}$, demonstrating a computational precision that is at least two orders of magnitude higher than that of FP32. This result implies that for most nonlinear materials with $\chi^2$ values below $10^{-6}$, transmittance spectrum simulations using FP32

become inherently unreliable. These findings highlight the critical role of numerical precision in computational science, particularly in fields where small numerical deviations can lead to substantial errors. As demonstrated in both DFT and FDTD simulations, the limitations of single precision introduce significant inaccuracies, especially in cases involving highly sensitive physical properties. This also highlights the necessity of carefully selecting numerical precision levels when conducting computational simulations, particularly in scientific machine-learning applications where maintaining the reliability of results is essential. While FP32 is adequate for various routine or weakly nonlinear calculations, our benchmarks show that in strongly nonlinear regimes it can introduce critical artifacts; the exact thresholds and representative case studies are provided in Appendix B.2.

# 3 Numerical precision issue in scientific ML

As demonstrated in the previous section, numerical precision can significantly affect the outcomes of traditional scientific simulations and potentially influence the results of scientific research. This naturally leads to an important question: **Do ML models designed for scientific tasks also suffer from similar precision-related issues?** To investigate this question, we survey various studies that apply ML to scientific research, searching for cases where the precision issue has been reported. We also conduct simple toy experiments to further assess the impact of numerical precision in ML-based scientific tasks. Through these analyses, we seek to determine whether the precision issue is a *significant challenge* or *just a theoretical concern*.

## 3.1 Impact on machine learning potential

The first example we present is an ML potential[1] [44, 50], which is closely related to Section 2.1. Fundamentally, ML potential models aim to compute potential energy and the associated forces for a given material structure, offering a much faster alternative to traditional quantum mechanical calculations. Due to their wide range of applications, ML potentials have been extensively studied not only in physics, materials science, chemistry, and biology but also within the ML community [14, 80, 93, 30, 90, 10, 9]. In addition, property prediction and generation for material or drug discovery have also been actively explored, making ML potentials a familiar subject for ML researchers. In our position paper, we focus specifically on neural network potentials, a class of ML potentials built on neural architectures. Since ML research often treats energy and force values in the same manner as other material properties, our discussion extends naturally to broader property prediction tasks.

A key challenge in ML potential studies lies in effectively representing and processing atomic information in three-dimensional space while ensuring rotational and translational equivariance or invariance. To tackle this, the field has evolved from vanilla graph neural networks [88] and transformers [106] to more specialized architectures that satisfy these constraints, achieving higher prediction accuracy [90, 28, 86, 10, 9, 27, 100, 54, 26]. As a result, many recent models are now integrated into widely used libraries or simulation software, such as the Atomic Simulation Environment (ASE) [5, 52] and LAMMPS [77, 101]. This demonstrates that ML potential models are increasingly employed in practical research; thus, any numerical precision issues arising in these models could have significant implications for scientific discoveries.

Consequently, we aimed to investigate whether existing ML potential models suffer from precision issues. To this end, we surveyed the pretrained checkpoints of various ML potential models available in the ASE library to determine whether they support FP64 precision. Interestingly, among several models, only MACE [9] provides pretrained checkpoints trained in FP64, while other models appear not to have considered FP64 training. Even before detailed analysis, this observation suggests that the ML potential community may not be fully aware of the potential significance of numerical precision.

To preliminarily understand the effect of precision, we conducted two toy experiment using MACE, the only model that provides FP64-trained parameters. In the first experiment, we examined the impact of numerical precision on the accuracy of potential energy surface (PES) reconstruction. We selected an ethanol molecule as a representative small organic system containing multiple atom types (C, H, and O). Then, by moving one of its carbon atoms along a certain path, we observed changes in the potential energy and forces. We compared FP32 and FP64 predictions by applying the built-in type conversion in the MACE code to the FP64-trained checkpoint. The results indicate that the

---

[1]In other domains, the term *machine learning interatomic potential* (MLIP) is also used.

differences between FP32 and FP64 remain within approximately 1 meV for energy and 0.02 meV/Å for force, which are margins typically considered acceptable in small-molecule simulations. Further experimental details can be found in Appendix B.3 and Figure 4.

Afterwards, we turned on a more advanced task, predicting vibrational properties, which contains richer information than the PES itself. To make the setting more realistic, we computed the vibrational modes of the *oseltamivir* molecule, which is the active ingredient of the anti-influenza drug *Tamiflu*. Table 2 summarize a subset of the calculated vibrational-mode frequencies; for modes 10 and 11, the discrepancies reach about 1.7 and 1.4 $cm^{-1}$, respectively. The differences up to 1.7 $cm^{-1}$ seems not significant, but this value may affect huge influence when analyzing the vibrational mode from the measured data. In general, the spectral resolution of Raman and infrared spectroscopy instruments, which generally utilized to measure the vibrational properties of materials, is varied from few $cm^{-1}$ for 10,000-50,000 USD to sub $cm^{-1}$ for over 100,000 USD. These comparable spectral resolutions of real-world instruments suggest that researchers may encounter ambiguous cases when interpreting marginal values of certain vibrational modes. For instance, consider a scenario where a researcher obtain a measured vibrational frequency of 78.2 $cm^{-1}$ for the oseltamivir molecule. Which vibrational mode should be assigned to this value? Calculations performed using FP32 precision would suggest mode 10, with a difference of only 0.36 $cm^{-1}$. Conversely, FP64 precision calculations would favor assignment to mode 11, despite the slightly larger difference of 0.59 $cm^{-1}$.

The aforementioned results suggest that while FP32 calculations may suffice for tasks such as molecular dynamics simulations based solely on PES, they may fall short when predicting more sensitive physical properties such as vibrational spectra. This emphasizes the importance of task-specific evaluation and precision-aware analysis, particularly when moving beyond PES-level predictions toward richer, experimentally comparable observables.

Nevertheless, our experimental framework was intentionally simplified, and these findings should not be overinterpreted as definitive evidence regarding machine learning models' sensitivity or insensitivity to numerical precision variations. Furthermore, the FP32 model evaluated in this study was initially trained using FP64 precision and subsequently converted to FP32 for inference purposes. A model trained exclusively in FP32 from initialization could exhibit different behavior. In fact, Batatia et al. [8] report that NequIP [10] exhibits different numerical sensitivity when trained in FP32 versus FP64, and Maxson et al. [62] also discuss similar issues. These observations highlight the importance of carefully assessing numerical precision in ML potential models and the need for systematic benchmarks regarding precision.

## 3.2 Impact on physics-informed neural network

Beyond the fundamental equations mentioned in the previous section, various subfields of natural science describe natural phenomena using differential equations. For example, in fluid dynamics, including weather prediction, Navier-Stokes, continuity, and heat transfer equations are used [105, 11]. Moreover, differential equations such as the Black-Scholes equation [17] are also employed in fields beyond natural sciences, such as financial engineering. Many of these equations either lack general analytical solutions or are too complex to be solved manually. As a result, numerical methods have been developed over time, leading to techniques such as the Euler method, Runge-Kutta methods, and Picard method [20, 96]. These techniques have also influenced modern approaches in ML, including diffusion models, NeuralODEs, and deep equilibrium models (DEQs) [35, 95, 21, 6].

The concept of the PINNs [82] leverages automatic differentiation (autograd), fundamental to backpropagation, to solve differential equations using neural networks. Due to its simple yet powerful approach, PINNs have been widely adopted in scientific domains that rely on numerical methods. This section explores whether numerical precision issues also arise in PINNs and investigates related challenges through a literature survey.

First, Nakamura et al. [68] explicitly discussed the impact of numerical precision in scientific research, reporting that training PINNs with FP32 failed, whereas FP64 did not: *from a comprehensive standpoint, FP32 computation has a risk of failure for the present problem compared with FP64.* This work applies PINNs to a specific fluid dynamics problem involving surface tension modeling, which requires up to fourth-order derivatives, making it a specialized case of differential equations. Although this is a specific scenario, it is a real-world scientific study, demonstrating that precision issues can significantly impact the practical use of PINNs.

Meanwhile, Sharma and Shankar [91] were well aware of precision issues and leveraged this understanding to improve the methodology of PINNs. The key idea of their work is to replace certain autograd operations in PINNs with a specialized finite difference method, reducing the computational cost associated with autograd. Here, to compensate for the loss of accuracy introduced by finite difference approximations, the authors proposed using high-precision (FP64) training. As a result, the reduction in computational cost from bypassing autograd exceeds the overhead introduced by FP64 operations, leading to an overall speedup that makes their approach faster than a vanilla PINN in FP32. Beyond the fields of PINNs and scientific ML, this study introduces a novel perspective on utilizing high-precision models in neural network research.

Thus, in the context of PINNs, a comprehensive study is needed to systematically assess the impact of numerical precision issues on scientific research. Fortunately, many fields share similar types of differential equations, *e.g.*, Laplace equation in electrostatics and fluid dynamics, where it describes electric potential distribution and velocity potential in inviscid flow, respectively. By focusing on the precision challenges of commonly used differential equations and rigorously validating PINNs in this context, such research could have a substantial impact across multiple domains.

## 3.3 Challenges for large language models

The emergence of LLMs in scientific applications is accelerating, further raising concerns about numerical precision in such domains. To investigate these concerns, we examine both existing studies and empirical evidence that highlight precision-related challenges in LLM applications. The integration of LLMs in scientific domains follows two distinct approaches. The first involves direct inference without architectural modifications, where scientific data is transformed into natural language format for existing LLM architectures [84, 37, 59]. The second approach develops specialized architectures that combine domain-specific encoders with fine-tuned language models, preserving the intrinsic properties of scientific data while leveraging LLM capabilities [53, 74].

Regarding the first approach, unlike conventional scientific models, LLMs generate outputs based on tokens, which may compromise prediction accuracy. Numerous studies have demonstrated that LLMs struggle with symbolic tasks [110, 114], similar to their difficulties in numerical predictions. For instance, these models often fail to accurately count the occurrences of specific characters within words (*e.g.*, counting the letter 'r' in 'strawberry') or comparing the size of decimal numbers (*e.g.*, determining whether 3.9 is larger than 3.11[2]). This limitation stems from their fundamental architecture, where words are processed as sequences of tokens rather than as individual alphabetic characters or numbers. Although various studies [110, 46, 114, 15] have been proposed to address these challenges, symbolic manipulation remains a significant obstacle for LLMs. Consequently, their application in scientific tasks requires careful consideration and validation.

Another critical consideration in LLM deployment is the continuous increase in model size. For instance, the open-source Llama series demonstrates this trend clearly: LLaMA (65B parameters) grew to Llama-2 (70B) and further to Llama-3.1 (405B) [103, 104, 60], and more recently, DeepSeek-v3 has pushed this expansion even further, reaching 671B [23]. Such explosive growth in model sizes across LLMs has resulted in a substantial increase in computational costs for both training and inference. To mitigate the budget, researchers commonly employ parameter quantization techniques by reducing model precision to lower-bit formats [57, 25, 58], sometimes even 1-bit representations [112].

However, these optimization strategies fundamentally conflict with the stringent precision requirements of scientific computing applications, as emphasized throughout our analysis. This issue is particularly critical for the second approach, where domain-specific encoders, which are often derived from scientific ML models, serve as feature extractors. If quantization significantly reduces the precision of the extracted features, the LLM may fail to process them accurately, potentially degrading overall model performance. For example, Li et al. [53] employed UniMol [121], a model broadly categorized as an ML potential, as an encoder. Even if the encoder provides highly precise features, the LLM's lower precision representations may obscure this information, leading to inaccurate final predictions. This inherent trade-off between computational efficiency and numerical precision highlights the necessity of careful consideration when integrating LLM into scientific applications.

---

[2]Recent large language models exhibit systematic errors in decimal comparison due to tokenization artifacts. When comparing 3.9 and 3.11, models tokenize these as ['3', '.', '9'] and ['3', '.', '11'] respectively, leading to incorrect digit-wise comparison (9 vs. 11) rather than proper decimal evaluation. As of early 2025, while GPT-4 has resolved this specific case, Claude 3.7 continues to incorrectly identify 3.11 as larger than 3.9.

## 4    Suggestions for Advancing Scientific ML

Building upon previous discussions, we present key suggestions for the scientific ML community.

**Benchmarking and reporting FP32 vs. FP64 results**    Scientific ML typically necessitate higher precision than general ML tasks to ensure reliability. While predictive accuracy is the primary focus, other factors such as training time, inference latency, and energy consumption remain significant constraints. Consequently, researchers should explicitly report the numerical precision used in their studies, conduct comparative analyses between the implementations of FP32 and FP64 where applicable, and publicly release FP64-trained models to improve reproducibility and facilitate collaborative research. To support meaningful evaluations, standardized benchmarks that capture precision sensitivity across diverse scientific tasks are essential. Such benchmarks would provide a consistent framework for quantifying trade-offs between numerical precision, computational efficiency, and reproducibility in scientific ML research.

**Exploring high-precision models and mixed high-precision training**    Inspired by mixed-precision training [65], we propose extending this concept to high-precision training by identifying precision-sensitive layers and selectively training them using FP64 arithmetic. This approach mirrors conventional mixed-precision strategies that utilize reduced precision (*e.g.*, FP32 and FP16) for most network layers while maintaining higher precision for numerically sensitive operations such as batch normalization and softmax. This direction holds significance from an energy efficiency perspective, as FP64 training inherently consumes more energy than FP32. While scientific ML offers computational advantages over traditional scientific computing methods, energy consumption remains a persistent concern. Investigating novel model architectures and training techniques that preserve high numerical precision while enhancing energy efficiency will be essential for broader adoption of scientific ML.

**Collaboration with natural scientists**    Achieving meaningful progress in scientific ML requires interdisciplinary collaboration with with natural scientists. This is not merely a conceptual argument but a practical requirement, as ML researchers often lack the domain-specific intuition to determine the appropriate level of numerical precision for a given scientific task. For instance, research on ML potential is published in both traditional scientific journals and ML conferences, yet the evaluation criteria and priorities differ substantially between these communities [9, 48]. Strengthening the collaboration will help bridge this gap, ensuring that precision requirements align with both scientific validity and practical usability.

**Integrating ML into traditional computational methods**    Rather than exclusively developing high-precision ML models, one of the alternative approaches is to integrate ML into traditional computational methods to achieve both accuracy and efficiency. One promising strategy is to employ ML models while acknowledging their inherent numerical limitations and using them to generate an approximate solution [3, 87, 69]. These ML-generated approximations subsequently serve as an initial guess for traditional computational methods, significantly accelerating convergence while preserving numerical precision. This hybrid approach presents a compelling solution for scientific applications where both computational speed and numerical accuracy are necessary.

## 5    Alternative Views

This section presents alternative views challenging our position and offers responses to these concerns.

**Q1: Is the extra computation cost due to higher precision tolerable?**    The most straightforward negative impact of using higher precision is the increased computational burden. For example, on NVIDIA A100 and H100 GPUs, FP64 operations are approximately twice as slow as FP32 operations. While this overhead may be acceptable for training that takes only a few hours, it becomes prohibitive for large-scale models trained across multiple GPUs over longer periods spanning several weeks or months. In addition, certain classes of GPUs (*e.g.*, RTX A6000) feature intentionally constrained FP64 performance, with throughput ranging from 1/32 to 1/64 of their FP32 capabilities. This hardware limitation makes consistent development of scientific ML models using double precision computationally inefficient. Therefore, as discussed in Section 4, a systematic analysis of numerical

precision's impact on model accuracy becomes essential, enabling practitioners to selectively optimize the balance between accuracy and computational efficiency.

**Q2: Is the issue really about numerical precision, or could it be a capacity limitation of the model?**  An alternative perspective suggests that observed inaccuracies originate from fundamental limitations in network architecture or training methodologies rather than numerical precision constraints. This viewpoint posits that neural networks may lack sufficient expressivity to solve a given task, regardless of precision considerations. To distinguish numerical issues from capacity concerns, we can employ numerical analysis tools including condition numbers and numerical sensitivity analysis (*e.g.*, interval arithmetic [33]), to determine whether errors arise from numerical instability. Since modern neural networks heavily rely on matrix operations, existing research on matrix sensitivity provides a robust analytical foundation. These insights can help clarify the relationship between numerical stability and model expressivity.

**Q3: If certain scientific computing tasks are not sensitive to numerical precision, does it matter?**  While not all scientific tasks require high numerical precision, focus should be directed toward fields where high precision is essential, such as quantum chemistry, materials science, and nonlinear physics, where even slight inaccuracies can lead to significant deviations. Currently, there is still limited understanding of which tasks, models, and environments are most affected by numerical precision and what factors contribute to these sensitivities. A systematic analysis is necessary to identify precision-critical cases before making broad assumptions about acceptable precision levels. Until a clear understanding is established, a precision-aware approach should be considered, while relaxed conditions can be applied only to tasks that demonstrably insensitive to precision.

Certain scientific tasks may not require explicit consideration of numerical precision, particularly those where logical reasoning is more critical than numerical accuracy, such as tasks relying on LLMs. These include explaining or summarizing experimental results or literature [111], generating hypotheses for scientific research [49, 61], providing guidance for tasks where the methodology is not clearly defined (*e.g.*, retrosynthesis), and assisting scientific educations [16]. In such cases, the role of ML extends beyond numerical fidelity, emphasizing knowledge synthesis and interpretability.

**Q4: Is it possible to design models that can avoid precision-related issues?**  Numerical instability in scientific computing frequently originates from precision-sensitive operations including numerical differentiation, integration, and eigendecomposition. Designing scientific ML models that avoid these operations and instead directly predict their outcomes can help mitigate such instability. ML potentials exemplify this approach by directly predicting energies from atomic structures, bypassing the numerically sensitive integration and eigendecomposition required in DFT. This perspective extends to examining whether individual neural network layers are numerically stable, similar to spectral normalization [66] which enforces Lipschitz continuity to stabilize training.

However, avoiding numerical instability through model design is not always practical, as scientists require understanding of underlying processes rather than just final predictions. This has led to ML models that mimic traditional scientific computations, such as NeuralODEs and DEQs [21, 6, 108], which explicitly model computational processes and align better with scientific domains emphasizing interpretability. While end-to-end approaches remain effective when predictive accuracy is the primary concern, many scientific domains continue to depend on numerical precision and computational understanding, making precision-related issues an important ongoing research area.

# 6  Conclusions

Scientific ML has become a major field in modern ML research, with the goal of developing models that contribute to scientific discovery. This position paper highlights the impact of precision issues, which can affect the practical usability of scientific ML models but have been largely overlooked. The precision issues in scientific ML are closely tied to ethical concerns regarding the reliability and explainability of scientific findings. In summary, our contribution lies in a practical step toward making scientific ML models more reliable, reducing the risk of misleading scientific insights due to numerical inaccuracies. If our simple yet easily actionable proposal becomes widely adopted in scientific ML research field, it can enhance the practicality and thereby accelerate scientific discovery.

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

# A  Details of computational methods

This section provides detailed information on the DFT and FDTD calculations.

## A.1  Density functional theory calculation

**Computational environment**  Quantum mechanical calculations were performed using PySCF version 2.7.0 [98]. To evaluate the impact of numerical precision, we conducted the same calculations using both single precision and double precision by declaring `np.float32` and `np.float64`, respectively. All simulations were conducted using two nodes of an AMD EPYC 7543 32-Core Processor.

**Simulation setup**  The input water molecule consists of a single oxygen atom at (0.000000, 0.000000, 0.000000) and two hydrogen atoms at (0.757000, 0.586000, 0.000000) and (-0.757000, 0.586000, 0.000000), respectively. To compare the geometry optimization result of a water molecule based on different exchange functionals, we performed both Hartree-Fock calculations and density functional theory calculations using the B3LYP functional [13, 113]. For both methods, we employed the 6-311++G(d,p) basis set [2]. The default convergence tolerances for structural stabilization were set as follows: $|\Delta E| < 1.00 \times 10^{-6}$, RMS-Grad $< 3.00 \times 10^{-4}$, Max-Grad $< 4.50 \times 10^{-4}$, RMS-Disp $< 1.20 \times 10^{-3}$, and Max-Disp $< 1.80 \times 10^{-3}$.

## A.2  Finite-difference time-domain calculation

**Nonlinear material properties**  Kerr media were modeled with a second-order nonlinear susceptibility ($\chi^2$) ranging from $10^{-12}$ to $10^{-2}$ and the refractive index was set to 1.65 to mimic the conventional nonlinear materials like beta barium borate. The nonlinear polarization of the material was expressed as:

$$P = \epsilon_0(\chi^{(1)}E + \chi^{(2)}E^2 + \chi^{(3)}E^3 + ...) \tag{1}$$

And the second-order nonlinear polarization term is represented as: $P^{(2)} = \epsilon_0\chi^{(2)}E^2$. Meep incorporates such nonlinear polarization terms into Maxwell's equations to simulate interactions between electromagnetic waves and the material in the time domain

$$\bigtriangledown \times H = \epsilon_0\frac{\partial E}{\partial t} + \frac{\partial P}{\partial t} \tag{2}$$

$$\bigtriangledown \times E = -\mu_0\frac{\partial H}{\partial t} \tag{3}$$

**Simulation setup**  The simulation domain consisted of a 100 $\mu$m medium, a 1 $\mu$m thick boundary layer, and 2 $\mu$m buffer regions at both ends. The spatial resolution was user-defined to capture fine electromagnetic field characteristics. Kerr media were placed at the center of the domain, with $\chi^2$ explicitly defined. The calculations were performed using Meep v1.29.0 [72], an open-source FDTD software, with both FP32 and FP64 precisions on a single core of an AMD Ryzen 5 8500G processor.

**Source and monitor definition**  The source was defined as a Gaussian plane wave with a central wavelength of 1.55 $\mu$m and a bandwidth of 0.15. Both the source and monitors were positioned 1 $\mu$m outside the nonlinear medium, with the electric field oscillating along the x-axis. Simulations were executed to allow sufficient decay of the fields after the source was turned off to confirm accurate measurements.

**Harmonic generation and analysis**  Using the Meep's add flux function, the optical flux outside the nonlinear medium was measured, and the transmitted power spectra of the fundamental frequency ($\omega$) and harmonic components ($2\omega$, $3\omega$) were calculated. The add flux function records the time-domain values of electric and magnetic fields at specific locations, then performs a Fourier transform to convert them into the frequency domain to compute flux. This process allows precise analysis of the intensity of each frequency component within the user-defined frequency range and intervals. The analysis frequency range extended from $\omega$/2 to 3.5$\omega$, encompassing all relevant frequency bands of interest. Flux measurements were particularly useful for understanding the interaction between

newly generated harmonic components and existing frequency components caused by the material's nonlinearity.

**Results and reproducibility**    Simulation results demonstrated how the intensity and distribution of harmonic components varied with changes in $\chi^2$. The nonlinear modeling capabilities of Meep enabled precise analysis of harmonic generation characteristics in nonlinear optical materials.

# B    Additional experimental results

In this section, we provide additional experimental results that supplement the results presented in the main text.

## B.1    Atomic coordination difference between FP32 and FP64

A detailed examination of the atomic coordinates in Table 1 further highlights the differences. While the coordinates obtained from FP64 differ only by approximately 0.01 Å for oxygen and hydrogen atoms, FP32 results display considerable deviation. Notably, the FP32-calculated atomic positions deviate by up to 0.4 Å from those obtained using FP64, a significant difference considering that the O-H bond length itself is only 0.957 Å. In addition, the total energy difference between FP32 and FP64 calculations is approximately 1.1 Hartree (equivalent to 29.93 eV), which exceeds the formation energy of water (2.9 eV) by more than an order of magnitude. This clearly indicates that the FP32 result corresponds to a structure that is impossible to exist in reality. These results demonstrate that FP32 lacks the numerical precision necessary to achieve sufficient convergence tolerance in scientific computations. The failure of a simple molecular system such as water to reach an optimized structure under FP32 precision indicates its fundamental limitations in scientific calculations.

Table 1: Comparison of atomic coordinates and total energy for geometry-optimized water molecule at FP32 and FP64 precision levels. Both calculations used the 6-311++G(d,p) basis set. As indicated by an asterisk (*), FP32 calculations failed to converge for both HF and DFT methods, while FP64 results show compatibility between HF and DFT.

| | | **HF** | | **DFT** (B3LYP) | |
| | | 6-311++G(d,p) | | 6-311++G(d,p) | |
| | | FP32 | FP64 | FP32 | FP64 |
|---|---|---|---|---|---|
| | $O_x$ | -0.000356* | 0.000000 | 0.009524* | 0.000000 |
| | $O_y$ | 0.246311* | 0.014028 | 0.578655* | 0.000780 |
| | $O_z$ | 0.000000* | 0.000000 | 0.000000* | 0.000000 |
| | $H_1 x$ | 0.453099* | 0.752792 | 0.026584* | 0.763642 |
| Atomic coordinates (Å) | $H_1 y$ | 0.534244* | 0.578999 | 0.998814* | 0.585902 |
| | $H_1 z$ | 0.000000* | 0.000000 | 0.000000* | 0.000000 |
| | $H_2 x$ | -0.453725* | -0.752792 | -0.024889* | -0.763642 |
| | $H_2 y$ | 0.534404* | 0.578999 | 0.998054* | 0.585902 |
| | $H_2 z$ | 0.000000* | 0.000000 | 0.000000* | 0.000000 |
| Total energy (Ha) | | -74.938* | -76.053 | *N/A*\* | -76.458 |

*Not Converged*

## B.2    Absorbed power density of a SiO$_2$ cylinder

To validate our FDTD workflow against a well-characterised linear system, we also simulated the absorption of a single silica (SiO$_2$) cylinder under normal-incidence plane-wave illumination.

**Geometry and material**    A two-dimensional square domain of side length 20 $\mu m$ was created, containing one infinitely long cylinder of radius 1.0 $\mu m$ centred at the origin. SiO$_2$ was taken from the built-in SiO$_2$ material in Meep, so its frequency-dependent permittivity—and hence refractive index—were implicitly evaluated at the simulation's centre frequency. Vacuum surrounded the cylinder. Mirror symmetry along the $y$-axis halved the computational cost.

**Source definition**     A continuous-wave Gaussian plane wave, polarized along $\hat{z}$ (out-of-plane $E_z$), impinged from the left boundary. The central wavelength was 1.0 $\mu m$ with a 10 % fractional bandwidth, wide enough to sample the vicinity of the design wavelength while narrow enough to approximate monochromatic excitation.

**Monitors and post-processing**

- **Incident-flux monitor.** A line segment at $x = -2.0\mu m$ recorded the power of the incoming wave, serving as a reference for normalising absorption.

- **Absorbed-flux box.** A closed rectangular contour wrapped tightly around the cylinder. By integrating the Poynting vector over this surface, net absorbed power $P_{\text{abs}}$ was obtained directly.

- **DFT field monitor.** A square region coincident with the flux box captured $E_z$ and $D_z$ fields in the frequency domain via Meep's discrete Fourier transform facility. Absorbed power density was then evaluated volumetrically as

$$p_{\text{abs}}(f) = 2\pi f \, \text{Im}\big(\overline{E_z} \, D_z\big) = 2\pi f \, \text{Im}(\varepsilon) \, |E_z|^2.$$

   where $f$ is frequency, $\varepsilon$ is the complex permittivity of SiO$_2$, and the overbar denotes complex conjugation. Integrating $p_{\text{abs}}$ over the cylinder volume reproduced $P_{\text{abs}}$ obtained from the flux box, providing a cross-check on numerical consistency.

**Simulation parameters**     The same spatial resolution used in the nonlinear study (*i.e.* user-defined) was retained to capture sub-wavelength field variations near the curved surface. Perfectly matched layers of 2 $\mu m$ thickness enclosed the domain to eliminate spurious reflections.

This linear-dielectric benchmark served two purposes: (*i*) it confirmed the accuracy of our absorption-post-processing pipeline before applying it to nonlinear scenarios, and (*ii*) it provided a reference scale against which to compare the additional harmonic-generation pathways introduced by the Kerr media.

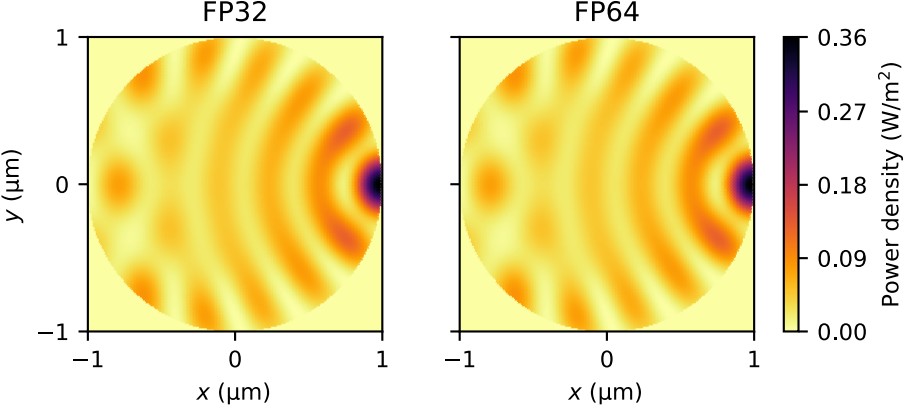

Figure 3: Absorbed power density of the SiO$_2$ cylinder computed with FP32 and FP64.

**Results and discussion**     In double precision (FP64) the spatially averaged absorbed–power density over the silica cylinder is $\langle p_{\text{abs}}\rangle = 3.2475 \times 10^{-2}$ $(W/m^2)$, with a standard deviation $\sigma = 3.4241 \times 10^{-2}$. Using single precision (FP32) we obtained $\langle p_{\text{abs}}\rangle = 3.2476 \times 10^{-2}$ and $\sigma = 3.4243 \times 10^{-2}$. The absolute differences between the two datasets are, respectively, $\Delta\langle p_{\text{abs}}\rangle = 1.96 \times 10^{-5}$ and $\Delta\sigma = 3.02 \times 10^{-5}$, which correspond to relative errors of $6.0 \times 10^{-2}$ % and $8.8 \times 10^{-2}$ %. These discrepancies are two orders of magnitude smaller than the intrinsic statistical spread of the fields and therefore negligible for any practical analysis.

This near-identity arises because (*i*) SiO$_2$ is essentially loss-free at the operating wavelength, so the dynamic range of $E_z$ and $D_z$ is modest; and (*ii*) the perfectly matched layers efficiently remove outgoing waves, preventing late-time reflections that could amplify numerical noise.

We therefore conclude that, for linear absorption in a dielectric cylinder, Meep's single-precision kernel yields numerically indistinguishable results from double precision while offering lower memory usage and faster runtimes. This benchmark justifies the use of FP32 for the more demanding nonlinear Kerr-media simulations reported in the main text.

### B.3 Toy experiments using MACE potential

**Experimental setup** Both toy experiments described in Section 3.1 were conducted under the same experimental setup. Among the various MACE model families, we used the `MACE-OFF23 medium` checkpoint [47]; the MACE codebase version was 0.3.10, and the ASE library [5, 52] version was 3.24.0. Each experiment was completed within 10 minutes on a single NVIDIA H100 GPU.

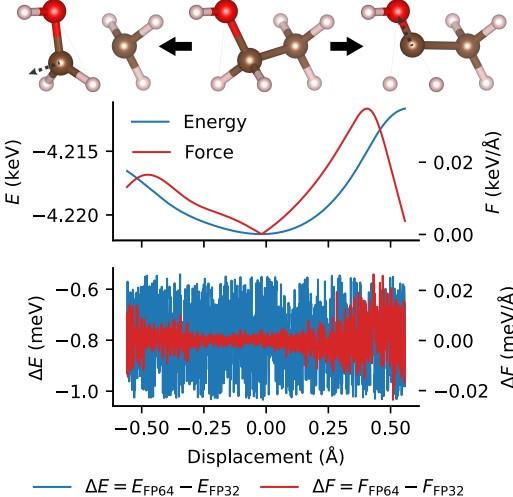

Figure 4: MACE model calculations showing energy (red) and force (blue) changes during carbon atom displacement in ethanol. FP32 versus FP64 precision reveals minimal deviations (1 meV energy, 0.02 meV/Å force).

**Ethanol experiment** We selected one of the carbon atoms in the ethanol molecule, specifically the one closest to the oxygen atom, and displaced it along a predefined direction while computing the corresponding energies and forces. The top panel of Figure 4 visualizes this setup. The original ethanol structure is shown in the center, with the displaced structures placed on either side. The direction of the carbon atom's movement is indicated by a dotted arrow.

The PES resulting from this displacement is shown in the middle panel. The energy (left y-axis) reaches a minimum when the displacement is zero, increases as the carbon atom moves away from its original position due to growing structural deformation, and then decreases again. The force (right y-axis) exhibits a similar trend, increasing from near zero and then decreasing as the displacement reverses.

The bottom panel shows the differences between FP32 and FP64 predictions for energy and force along the displacement path. The energy difference remains relatively uniform across the range. In contrast, the force difference is minimal near the equilibrium structure and increases gradually as the displacement moves the system farther from the optimal configuration.

**Oseltamivir experiment** We downloaded the 3D structure of oseltamivir from PubChem[3] and used it in our experiment. To compute the vibrational modes of a molecule, structural optimization is required prior to vibrational analysis. For this purpose, we performed geometry optimization using the MACE-OFF model and the ASE library, employing the BFGS optimizer with the maximum force threshold set to 0.01 eV/Å.

---

[3] https://pubchem.ncbi.nlm.nih.gov/compound/Oseltamivir

Table 2: Calculated vibrational modes of the oseltamivir molecule, the active ingredient in Tamiflu, using the MACE framework. Out of 30 total modes, four imaginary phonon modes were excluded, and only the 26 real phonon modes are presented.

| Mode | Frequency (cm$^{-1}$) | | |
| --- | --- | --- | --- |
| | FP32 | FP64 | Δ |
| 1 | 0.418 | 0.060 | 0.357 |
| 2 | 2.003 | 1.403 | 0.600 |
| 3 | 20.729 | 21.107 | 0.377 |
| 4 | 23.581 | 23.481 | 0.099 |
| 5 | 30.129 | 30.058 | 0.071 |
| 6 | 32.152 | 31.345 | 0.807 |
| 7 | 42.711 | 42.550 | 0.161 |
| 8 | 59.571 | 58.907 | 0.663 |
| 9 | 60.734 | 60.132 | 0.602 |
| **10** | **77.851** | **76.113** | **1.738** |
| **11** | **80.128** | **78.696** | **1.431** |
| 12 | 92.388 | 92.069 | 0.318 |
| 13 | 94.896 | 94.720 | 0.176 |
| 14 | 103.763 | 103.375 | 0.387 |
| 15 | 113.600 | 113.514 | 0.085 |
| 16 | 144.647 | 144.908 | 0.260 |
| 17 | 153.184 | 153.488 | 0.304 |
| 18 | 180.164 | 179.934 | 0.230 |
| 19 | 183.479 | 183.414 | 0.065 |
| 20 | 212.501 | 212.516 | 0.014 |
| 21 | 228.214 | 228.076 | 0.138 |
| 22 | 240.207 | 240.086 | 0.121 |
| 23 | 250.314 | 250.124 | 0.190 |
| 24 | 255.417 | 256.290 | 0.873 |
| 25 | 263.003 | 262.807 | 0.195 |
| 26 | 270.000 | 269.875 | 0.124 |

Following the optimization, we computed the vibrational modes using the ASE library as well. A total of 30 modes were obtained, among which four modes were identified as imaginary phonon modes. The remaining 26 real phonon modes are reported in Table 2.

