# OpenReview forum: "Position: Significant impact of numerical precision in scientific machine learning"
_NeurIPS.cc/2025/Position_Paper_Track — Submitted to NeurIPS 2025 Position Paper Track_

### Official Review · Reviewer_99x3 · 2025-07-26

**Significance:** 2
**Presentation:** 2
**Rating:** 4
**Confidence:** 4

**Summary:**

The paper argues that numeric precision is a critical aspect of scientific computing, and therefore should be a more explicit focus of attention in scientific machine learning. The paper discusses examples from learned potentials, physics informed neural nets, and the use of llm-based models in science. The paper calls for a rigorous study of the impact of numeric precision on the accuracy of ML models used in various scientific applications.

**Strengths:**

The paper tries to bridge gaps between requirements in the scientific computing community and the ML community and broaden the dialogue around the differing requirements of scientific computing. The paper gives concrete examples of scientific computing methods that are impacted by numeric precision issues. A good case is made that in certain cases, numeric issues can lead to poor performances in scientific applications, and the paper provides a clear call for action for studying the phenomenon more closely.

**Weaknesses:**

The paper does not provide strong evidence that numeric precision is an issue in any of the three areas that are investigated. In particular, for section 3.1, the mismatch between the 32bit and 64bit models does not necessarily indicate that the 64bit model is more accurate.

While precision might impact accuracy, it does not seem fundamentally different than any of the other hyper-parameter that need to be adjusted to obtain good performance, such as learning rate, weight decay, architecture choices etc. If there is a qualitative difference, that should be made more clear in the paper.

There is also not a strong argument about what can be gained from a systematic review. A thorough review will likely yield the answer of "it depends". It might be possible to specify in which cases precision is critical (such as in the case mentioned in 3.2), which would be useful for practitioners. However, this would be a case of "beneficial guidance" more than "significant impact".

**Questions:**

Can you elaborate on how precision issues in scientific ML are closely tied to ethical concerns regarting the reliability of the findings? Generally, accuracy of ML methods is only statistical in nature, and is measured usually (in supervised learning) using a hold-out test-set. This measurement of reliability is unrelated to the precision or reliability of the model. The reliability of any model can usually only be assessed in this way.

In Q2, you mention "observed inaccuracies". Can you be more specific in what you mean by that? Also, are you attributing inaccuracies to precision issues? It is quite rate that inaccuracies of a model can be directly attributed to any particular aspect of a model.

**Alternative Position:**

Yes, and alternative positions are well-considered and named but not addressed

**Author Identification:**

No.

**Context:**

3

**Discussion:**

2

**Ethics:**

["NO or VERY MINOR ethics concerns only"]

**Position:**

Yes, the paper argues for or against a position related to machine learning.

**Support:**

2

**Thoroughness:**

4

---

### Official Review · Reviewer_sxo2 · 2025-07-28

**Significance:** 3
**Presentation:** 3
**Rating:** 7
**Confidence:** 4

**Summary:**

This paper argues that numerical precision (FP64) is critical in scientific machine learning (ML), highlighting how insufficient precision (FP32) can significantly degrade outcomes. The authors advocate for systematic benchmarking of FP32 vs. FP64, encouraging the community to report and release FP64-trained models.

**Strengths:**

The paper clearly argues for the importance of numerical precision in scientific ML, supported by detailed and relevant examples from various scientific fields. Recommendations for systematic benchmarking and public release of FP64 models are actionable and likely to be impactful. Alternative viewpoints are thoroughly addressed, enhancing the robustness of the paper.

**Weaknesses:**

The discussion could benefit from a more detailed quantitative analysis of the computational overhead associated with FP64. Additional empirical evidence specifically from ML experiments would strengthen the argument further. Exploration of mixed-precision strategies is mentioned but could benefit from deeper technical elaboration.

**Questions:**

Could the authors provide preliminary data or further insights into the computational overhead involved in using mixed FP32-FP64 precision?

Have the authors identified scenarios where FP32 is sufficient, and could they elaborate on criteria that distinguish these scenarios from FP64-sensitive tasks?

**Alternative Position:**

Yes, and alternative positions are well-considered and addressed by the argument

**Author Identification:**

No.

**Context:**

4

**Discussion:**

3

**Ethics:**

["NO or VERY MINOR ethics concerns only"]

**Position:**

Yes, the paper argues for or against a position related to machine learning.

**Support:**

3

**Thoroughness:**

4

---

### Official Review · Reviewer_gX4x · 2025-08-12

**Significance:** 2
**Presentation:** 2
**Rating:** 3
**Confidence:** 4

**Summary:**

The papers argues that the precision of computations should be increased in the hope that more accurate results will be produced.

**Strengths:**

The authors are absolutely right (lines 4-6) that in "materials science, where even small numerical differences can lead to significant inaccuracies in physicochemical properties".

The much worse overlooked problem is the discontinuity of cell-based representations of periodic materials, because almost any perturbation of atoms can arbitrarily scale up a primitive (or reduced) cell, which has been know experimentally since 1965 (Lawton, Stephen L., and Robert A. Jacobson. The reduced cell and its crystallographic applications. No. IS-1141. Ames Lab., Iowa State Univ. of Science and Tech., US) and comparisons by unit cells (or motifs) unreliable as well as materials databases using unstable cells.

Figure 1 gives a crucial counter-example to the DFT outputting physically unrealistic molecules of water. The included example does not prove that the DFT is correct at any (lower or higher) precision except that this single example seems correct. In general, examples prove nothing, while counter-examples disprove conjectures, so Figure 1 has disproved the conjecture that the DFT can be used in practice.

**Weaknesses:**

Responding to "This need for high precision extends to the emerging field of machine learning for scientific tasks, yet it has not been thoroughly investigated" (lines 6-7), the precision have been thoroughly investigated by mathematicians and other scientists, which has led to the development of chaos theory discovered by Edward Lorenz using computers, so computer scientists should learn this.

Increasing precision is worthless because any real (not too simple) dynamical system unpredictably changes its behavior under tiny perturbations of initial conditions or parameters. This "butterfly effect" emerges even for 1-variable maps: x->4x(1-x), where perturbing any initial x in [0,1] produces a random sequence.

The chaos theory was honored by the 1977 Nobel Prize in Chemistry to Ilya Prigogin but seems to be completely forgotten in favor of artificial illusions consuming more and more resources by promising to quantify uncertainties that should be embraced in the real world.

**Questions:**

Can the authors list machine learning algorithms that are guaranteed to produce stable results, e.g. Lipschitz continuous under perturbations? Such a list would provide a good incentive not to waste time and resources on impractical approaches, which endanger the life on Earth, at least through increased emissions of greenhouse gases.

**Alternative Position:**

Yes, and alternative positions are trivial straw-man arguments

**Author Identification:**

No.

**Context:**

1

**Details Of Ethics Concerns:**

Following the guidelines below, the proposed but unjustified increase of precision will likely consume even more precious resources (electricity, water, etc) and will accelerate carbon emissions endangering the life on Earth.

Environment: Researchers should consider whether their research is going to negatively impact the environment by, e.g., promoting fossil fuel extraction, increasing societal consumption or producing substantial amounts of greenhouse gases.

**Discussion:**

2

**Ethics:**

["Major Concern: Environmental impact"]

**Position:**

Yes, the paper argues for or against a position related to machine learning.

**Support:**

1

**Thoroughness:**

3

---

### Note · Authors · 2025-09-05

**1-11 Submit Again:**

Unsure

**1-1 Submission Process:**

4

**1-3 Future Development:**

This year, it appears that reviewers for the main track and position track were recruited separately. For next year, we suggest an alternative approach: rather than recruiting reviewers separately, position papers could be assigned to reviewers who are randomly sampled from the pool of main track reviewers. For example, among main track reviewers who are originally expected to review N main track papers, some could instead be assigned to review (N - 1) main track papers and one position track paper.

**1-4 Interest:**

["Structured debates on controversial topics", "Workshops for developing position papers"]

**1-5 Thoughtful:**

6

**1-6 Supportive:**

5

**1-7 Technical Aspects Versus Position:**

8

**1-8 Gate Keeping:**

4

**1-9 Camera Ready Changes:**

First, we will incorporate the reviewers' feedback by revising and supplementing the content, particularly in Sections 4 and 5. In addition, we found additional papers that support our position, and we plan to incorporate these papers into the camera-ready version.

**3-1 Review Response1:**

gX4x

**3-2 Reaction To Review1:**

We thank the reviewer for raising interesting points that we had not previously considered. Although the reviewer strongly disagrees with our position, we believe that several of the concerns stem from misunderstandings of our claims. We therefore clarify these points below, with the hope that our responses will resolve the misinterpretations.

[Environmental concern]
First and foremost, we wish to clarify that our paper does not advocate for the unconditional use of higher precision. Instead, we argue for a systematic analysis of precision's impact to enable the selective and appropriate use of different precision levels for a given task. We made this consideration with full awareness of the computational cost increases from using higher precision and the resulting environmental concerns, as the reviewer pointed out. We discussed this aspect in Sections 4 and 5. We will explicitly emphasize environmental concerns in the camera-ready version.
Furthermore, while it is true that higher-precision computations increase energy costs, it is equally important to consider that lower precision may yield incorrect scientific decisions. This could lead to significant resource waste if researchers repeat experiments or even correct misguided scientific decisions.

[Chaos theory]
We agree with the reviewer's opinion that fundamental uncertainty exists in situations where chaos theory applies. However, we disagree with the reviewer's assertion that "increasing precision is worthless" based on chaos theory, and we would like to present two reasons for this position.
First, chaos theory is not a universal theory applicable to all scientific problems, but rather a theory applicable to dynamic systems that satisfy specific conditions.
Second, what chaos theory addresses is the intrinsic predictability of dynamic systems, while what our paper discusses is the stability according to numerical precision when predicting the state of any physical system using computers and ML models.

**3-3 Review Response2:**

sxo2

**3-4 Reaction To Review2:**

We sincerely appreciate reviewer sxo2’s positive and constructive feedback. Below we provide detailed responses to the reviewer’s questions.

[Computational overhead associated with FP64]
Following the reviewer's suggestion to include analysis of computational overhead associated with FP64, we extended our toy experiment on vibrational modes calculation for the Oseltamivir molecule (Section 3.1, B.3., and Table 2) to additionally measure inference time. The results showed 24.099 seconds for FP32 and 26.559 seconds for FP64, representing approximately 10 % overhead, which is significantly lower than the theoretical expectation of 2x based on H100's official specifications mentioned in Section 5. This lower-than-expected overhead occurs because actual model training and inference processes involve additional operations such as data transfer between GPU cores and memory. While it is a limited toy experiment, it demonstrates the positive finding that computational cost increases from using higher precision may be less than theoretical expectations. If mixed precision approaches were employed, we would expect even smaller increases.

[Identified scenarios where FP32 is sufficient, and could they elaborate on criteria that distinguish these scenarios from FP64-sensitive tasks?]
First, the Burgers equation is widely used as a demonstration for introducing physics-informed neural network (PINN) concepts, and when we experimented with this equation, we observed no significant differences between FP32 and FP64 model results.
Second, we would like to clarify that this paper is a position paper originating from the authors’ empirical observations of precision-related issues during scientific machine learning (ML) model development. Therefore, rather than presenting rigorous and generalized rules covering all scenarios, we focused on raising awareness of the problem's importance and suggesting future research directions through toy experiments and literature surveys.

**3-5 Review Response3:**

99x3

**3-6 Reaction To Review3:**

We thank reviewer 99x3 for the valuable and supportive feedback. The reviewer's detailed comments have helped us clarify our paper's core arguments and strengthen our logical framework.

[FP64 accuracy claims]
We clarify that our paper does not assert FP64 models are universally superior. Rather, we demonstrate that substantial differences can emerge between FP32 and FP64 computations, supported by empirical experiments and literature analysis. Our core argument is that numerical precision represents an underexplored variable significantly impacting scientific machine learning (ML) outcomes.

[Numerical precision as hyperparameter]
Traditional hyperparameters such as learning rate and weight decay are actively tuned with established protocols. Numerical precision, however, has been largely overlooked in scientific ML applications, creating a gap between traditional scientific computing and scientific ML domain. Therefore, highlighting that numerical precision can be an important variable and proposing its active consideration in research processes constitutes a meaningful contribution to bridging the paradigms of traditional scientific computing and scientific ML.

[Systematic analysis]
concerning the point that a systematic analysis would inevitably conclude that the importance of precision “it depends,” we argue that this observation precisely supports our position. Identifying the specific conditions under which “it depends” and clarifying these distinctions is an important scientific contribution. Currently, researchers in scientific ML lack formal guidance on when and why precision considerations become critical. The goal of systematic analysis is to provide such guidance, enabling researchers to make informed decisions about precision requirements for their specific applications.

---

### Meta-Review · Area_Chair_3j3j · 2025-08-29

**Rating:** 6
**Confidence:** 4

**Strengths:**

- The paper argues for a clear and relevant position, that the precision we use is relevant for the success of ML methods.
- It calls for benchmarking FP32 vs FP64 and the release of FP64-trained models, which is a concrete and actionable recommendation.
- The position is grounded from needs of the physics domain where insufficient precision can lead to real problems.

**Weaknesses:**

- The experiments that illustrate the need for high precision are rather suggestive and limited.
- The computational trade-offs are underexplored. Especially for foundation models, the computational burden of using FP64, as well as the environmental impact, would be substantial.
- The paper focusses on the precision of parameter representation, but this is just one of several factors, such as architectures, optimization method, data quality. The interplay between these factors is essential for understanding the need for higher precision, especially given the high computational burden it entails.

**Questions:**

-

**Ethics:**

The paper raises no ethical concerns.

**Thoroughness:**

3

---

### Decision · Program_Chairs · 2025-09-26

Reject